# Soluble Transferrin Receptor as Iron Deficiency Biomarker: Impact on Exercise Capacity in Heart Failure Patients

**DOI:** 10.3390/jpm13081282

**Published:** 2023-08-21

**Authors:** Maria del Mar Ras-Jiménez, Raúl Ramos-Polo, Josep Francesch Manzano, Miriam Corbella Santano, Herminio Morillas Climent, Núria Jose-Bazán, Santiago Jiménez-Marrero, Paloma Garcimartin Cerezo, Sergi Yun Viladomat, Pedro Moliner Borja, Blanca Torres Cardús, José Maria Verdú-Rotellar, Carles Diez-López, José González-Costello, Elena García-Romero, Fernando de Frutos Seminario, Laura Triguero-Llonch, Cristina Enjuanes Grau, Marta Tajes Orduña, Josep Comin-Colet

**Affiliations:** 1Bio-Heart Cardiovascular Diseases Research Group, Bellvitge Biomedical Research Institute (IDIBELL), 08907 L’Hospitalet de Llobregat, Spain; 2Community Heart Failure Program, Cardiology Department, Bellvitge University Hospital, 08907 L’Hospitalet de Llobregat, Spain; 3Department of Internal Medicine, Bellvitge University Hospital, 08907 L’Hospitalet de Llobregat, Spain; 4Cardiology Department, Bellvitge University Hospital, 08907 L’Hospitalet de Llobregat, Spain; 5Centro de Investigación Biomédica en Red de Enfermedades Cardiovasculares (CIBERCV), 28029 Madrid, Spain; 6Advanced Practice Nurses, Hospital del Mar, Parc de Salut Mar, 08003 Barcelona, Spain; 7Biomedical Research in Heart Diseases, IMIM (Hospital del Mar Medical Research Institute), 08003 Barcelona, Spain; 8Escuela Superior de Enfermería del Mar, Parc de Salut Mar, 08003 Barcelona, Spain; 9Primary Care Service Delta del Llobregat, Institut Català de la Salut, 08820 Barcelona, Spain; 10Primary Care Service Litoral, Institut Català de la Salut, 08023 Barcelona, Spain; 11Department of Medicine, Universitat Autònoma de Barcelona, 08193 Barcelona, Spain; 12Department of Medicine, Universitat Pompeu Fabra, 08002 Barcelona, Spain; 13Advanced Heart Failure and Heart Trasplant Program, Cardiology Department, Bellvitge University Hospital, 08907 L’Hospitalet de Llobregat, Spain; 14Department of Clinical Sciences, School of Medicine, University of Barcelona, 08036 Barcelona, Spain

**Keywords:** heart failure, iron deficiency, soluble transferrin receptor, submaximal functional capacity, quality of life, 6 min walking test (6MWT), Minnesota Living with Heart Failure Questionnaire (MLHFQ)

## Abstract

The soluble transferrin receptor (sTfR) is a marker of tissue iron status, which could indicate an increased iron demand at the tissue level. The impact of sTfR levels on functional capacity and quality of life (QoL) in non-anemic heart failure (HF) patients with otherwise normal systemic iron status has not been evaluated. We conducted an observational, prospective, cohort study of 1236 patients with chronic HF. We selected patients with normal hemoglobin levels and normal systemic iron status. Tissue iron deficiency (ID) was defined as levels of sTfR > 75th percentile (1.63 mg per L). The primary endpoints were the distance walked in the 6 min walking test (6MWT) and the overall summary score (OSS) of the Minnesota Living with Heart Failure Questionnaire (MLHFQ). The final study cohort consisted of 215 patients. Overall QoL was significantly worse (51 ± 27 vs. 39 ± 20, *p*-value = 0.006, respectively), and the 6 MWT distance was significantly worse in patients with tissue ID when compared to patients without tissue ID (206 ± 179 m vs. 314 ± 155, *p*-value < 0.0001, respectively). Higher sTfR levels, indicating increased iron demand, were associated with a shorter distance in the 6 MWT (standardized β = −0.249, *p* < 0.001) and a higher MLHFQ OSS (standardized β = 0.183, *p*-value = 0.008). In this study, we show that in patients with normal systemic iron parameters, higher levels of sTfR are strongly associated with an impaired submaximal exercise capacity and with worse QoL.

## 1. Introduction

Heart failure (HF) is a syndrome with a great impact on morbidity and mortality that portends an increasing burden on our healthcare system [1,2,3]. In this context, patient-reported outcomes, such as health-related quality of life (QoL), and functional outcomes, such as exercise capacity, are also greatly impaired and strongly correlate with clinical outcomes. Importantly, these two functional status measures are considered highly relevant outcomes from the patient’s and healthcare professional’s perspective [4,5,6].

Regarding this, many research efforts have been directed to understand the main drivers of impairment in exercise capacity and QoL. These have been carried out to provide new therapeutic options that would allow for advancing meaningful improvements of these functional measures [7,8,9,10].

In recent years, comorbidities like iron deficiency (ID) have been identified as potential modifiable factors that greatly influence functional status and QoL [3,11]. ID is estimated to be present in up to 50% of patients with HF. Several studies have confirmed that it is associated with decreased exercise capacity, QoL, and worse outcomes regardless of the left ventricular ejection fraction (LVEF) and hemoglobin (Hb) levels [12,13,14,15].

Several clinical trials have shown that intravenous iron replacement improves functional capacity and QoL in patients with HF regardless of the presence of anemia [8,9]. The definition for ID used in these studies is based on the blood concentration levels of ferritin and the % of transferrin saturation (TSAT) [8,9]. Thus, international clinical guidelines define ID as when the serum ferritin is below 100 nanograms per milliliter (ng/mL) or the TSAT is below 20 when the ferritin levels are between 100 and 299 ng/mL [3].

Then again, most experts suggest that iron deficiency should be conceived as a continuum from a normal iron status to overt systemic ID with a final impact on Hb levels, leading to iron-deficiency anemia [16].

The first stage of the transition along the continuum of iron status between these two poles would be a mild form of functional iron deficiency at the tissue level, with little impact on iron storage or the iron transport compartments [17]. In fact, the presence of ID in the heart is not associated with the classic definition of ID in the serum [18]. This early ID may only have a subtle impact on the functional status of patients, and thus, may be regarded as a type of subclinical ID state [16,17,19,20]. Considering this, biomarkers such as the soluble transferrin receptor (sTfR) have been suggested to allow for a better assessment of the iron demand at the tissue level and might better indicate the iron status.

Compared to TSAT (a marker of circulating iron availability) and ferritin (a surrogate marker of the amount of iron stored), the sTfR shows the best accuracy in predicting ID in bone marrow [21]. Catabolism and malnutrition can reduce serum transferrin disproportionately to serum iron, leading to elevations in TSAT. Several studies have shown that higher sTfR values are associated with mortality [21] and worse health-related QoL [22] in patients with HF.

However, these studies were based on patient cohorts with an elevated proportion of patients with overt systemic ID according to the ID definition suggested in clinical guidelines based on ferritin and TSAT [3]. Thus, the impact of subclinical ID defined as raised levels of sTfR, suggesting increased tissue iron demand, on the functional capacity and QoL in HF patients without overt systemic ID or anemia has not been properly explored.

Given the limitations mentioned above, our study aimed to explore the interplay among tissue ID, defined as higher sTfR values, QoL measured with the Minnesota Living with Heart Failure Questionnaire (MLHFQ), and submaximal exercise capacity using the 6 min walking test (6MWT) in a patient cohort with HF regardless of LVEF. Those with anemia and/or systemic ID defined as ferritin < 100 μg/L or ferritin 100–300 μg/L and TSAT < 20% were excluded.

## 2. Materials and Methods

The definition of the neuro-hormonal activation, myocardial function, genomic expression, and clinical outcomes in heart failure patients (DAMOCLES) study was a single-center, observational, prospective cohort study of 1236 consecutive patients diagnosed with HF recruited between January 2004 and January 2013.

Inclusion and exclusion criteria. The methodology of the DAMOCLES study was previously published by our group [13,14,22,23,24,25,26,27,28,29,30]. Briefly, for inclusion, patients had to be diagnosed with HF in accordance with the European Society of Cardiology diagnostic criteria, have at least one recent acute decompensation of HF requiring intravenous diuretic therapy (either hospitalized or in the day-care hospital), and be in stable condition at the time of study entry. The exclusion criteria were significant primary valvular disease, clinical signs of fluid overload, pericardial disease, restrictive cardiomyopathy, or hypertrophic cardiomyopathy, Hb levels < 8.5 g per deciliter (g/dL), active malignancy, and chronic liver disease. The study was approved by the local committee of ethics for clinical research and was conducted in accordance with the principles of the Declaration of Helsinki. All patients gave written informed consent before study entry.

For the present analysis, all DAMOCLES participants were considered for inclusion. Of them, we selected those patients that had a complete iron status evaluation, including sTfR values, normal hemoglobin levels (≥12 g/dL), and normal systemic iron status (serum iron > 33 micrograms per deciliter (µg/dL), ferritin > 100 ng/mL, and transferrin saturation > 20%). Those patients with missing baseline information relative to the iron parameters, QoL, or 6 MWT were excluded. Thus, the final cohort consisted of 215 patients.

ID definition based on sTfR. The purpose of the present study was to describe the association between sTfR, as a marker of increased iron demand and tissue iron deficiency, and the submaximal exercise capacity and QoL of non-anemic patients with HF and normal systemic iron status.

The primary endpoints were the distance walked in the 6 MWT and the overall summary score (OSS) of the self-administered MLHFQ, both upon inclusion in the study. The secondary endpoints included the proportion of patients with an impaired submaximal exercise capacity, defined as a 6 MWT distance < 300 m, and advanced New York Association (NYHA) class, defined as NYHA functional class III or IV at baseline. The scores of the physical, emotional, and social domains of the MLHFQ were also secondary outcomes.

Our hypothesis was that raised sTfR levels, indicating tissue iron deficiency, would be associated with an impaired submaximal exercise capacity and worse QoL.

Methods. A detailed baseline evaluation was performed for all participants at the study entry. This included the collection of information about demographic characteristics and compiling an exhaustive medical history to gather clinical and disease-related factors, such as NYHA functional class, co-morbidities, laboratory information, medical treatments, and the most recent LVEF. The sources of information were the medical history and standardized questionnaires.

sTfR levels were measured with the commercially available Beckman Coulter enzyme immunoassay (higher levels indicate increased iron demand). Tissue ID was defined as levels of sTfR > 75th percentile, which corresponds to 1.63 milligrams per liter (mg/L). We used this definition because there is no standardized or validated cutoff value for sTfR that defines tissue ID.

The submaximal exercise capacity was assessed using the 6MWT, based on the methodology previously published by our group [13].

Each patient performed a single 6 MWT using a standardized protocol as described in previous studies and guidelines [31]. The test was performed by a trained nurse who would ask the patient to walk the longest distance possible along a 30-m corridor in an interval of 6 min. The patient could stop or slow down at any time and then resume walking, depending on their degree of fatigue. The exercise test was stopped upon the patient’s request. The total distance covered in 6 min, early interruption of the test, and the presence of symptoms during the test were reported. To classify patients into 2 exercise capacity categories (impaired or preserved), a cutoff point of 300 m was chosen. This threshold has been reported previously as a predictor of mortality and morbidity in HF [32,33].

The details of the QoL evaluation in the DAMOCLES study have been previously reported [12,13,14,22,30]. In short, we used the MLHFQ. The MLHFQ is a self-administered disease-specific questionnaire for patients with HF. It comprises 21 items rated on 6-point Likert scales that represent the different degrees of impact of HF on QoL, from 0 (none) to 5 (very much). Therefore, higher scores indicate worse QoL. It provides a total score as well as scores for the different dimensions of QoL, namely the physical, emotional, and social. The MLHFQ has been translated into and validated in Spanish [34].

Statistical analysis. Using the baseline data from the DAMOCLES cohort, a cross-sectional descriptive analysis was performed. The demographic and clinical characteristics as well as the laboratory test results were summarized using basic descriptive statistics, both overall and categorized by sTfR levels, and divided into tertiles.

For the categorical variables, numbers and percentages are reported. The mean (standard deviation) or median (inter-quartile range) was used for the continuous variables, depending on the distribution of the variables. The χ^2^, Student’s T, and non-parametric tests were used to compare characteristics across strata.

Unadjusted generalized additive models (GAMs) were used to explore the parametric and non-parametric associations among the sTfR, 6 MWT distance, and QoL scores (overall and for every specific dimension). As a result of these analyses, smooth cubic spline curves of the estimated beta coefficients of risk of each event were plotted. The findings obtained in these unadjusted models were replicated in a multivariable GAM.

To further explore the associations between sTfR levels and tissue ID with the variables of the study (QoL scores and distance walked in the 6MWT), univariate and multivariate linear regression models were constructed. Additional univariate and multivariate binary regression models were constructed to evaluate the associations among sTfR levels, tissue ID, impaired exercise capacity, and advanced NYHA class. Backwards conditional stepwise methods were used for the binary regression models, whereas forward methods were used for the linear regression models. A collinearity analysis was performed, and the absence of tolerance < 0.3 (high collinearity) in the variables included in the final model was verified.

All multivariable models were adjusted by prognostic factors and determinants of HF severity (age, gender, systolic blood pressure, diabetes mellitus, LVEF, renal function, N-terminal pro-brain natriuretic peptide (NTproBNP) levels, etiology of HF, use of disease-modifying drugs, body mass index (BMI), hemoglobin, C-reactive protein, NYHA functional class, and comorbidity burden).

All statistical tests and confidence intervals (CI) were constructed with a type I error alpha level of 5%, with no adjustments for multiplicity. *p* values below 0.05 were considered statistically significant. All analyses were performed using the SPSS software package (version 22.0; IBM, Armonk, NY, USA) and the R software package (version 4.2.1; R Foundation for Statistical Computing, Vienna, Austria).

## 3. Results

A total of 1236 HF patients were included in the DAMOCLES study. However, the final cohort of this study consisted of 215 patients, as those with anemia and iron deficiency were excluded (Appendix A).

### 3.1. Baseline Patient Characteristics

The baseline characteristics of the study sample, both overall and according to the sTfR tertiles (lower tertile: sTfR < 1.11 mg/L, middle tertile: sTfR from 1.11 mg/L to 1.45 mg/L, and higher tertile: sTfR ≥ 1.46 mg/L), are presented in Table 1. The mean age was 70 ± 12 years, the mean LVEF was 43 ± 15%, and 62 patients (29%) were women. The mean sTfR values were 1.42 ± 0.66 mg/L, and tissue ID, defined as levels of sTfR > 75th percentile (corresponding to sTfR levels above 1.63 mg/L), was present in 54 patients (26%).

There were no differences among the groups in terms of age, LVEF, etiology of HF, or the neurohormonal treatment received (all *p*-value > 0.05). Patients with the highest sTfR tertiles, indicating tissue iron deficiency, showed a higher baseline heart rate (*p* = 0.020), worse functional class (*p* = 0.033), and slightly worse renal function (*p* = 0.024) when compared to the other sTfR categories.

### 3.2. Association of sTfR with Exercise Capacity and NYHA Class

An impaired submaximal exercise capacity, defined as a 6 MWT distance < 300 m, and an advanced NYHA class, defined as NYHA functional class III or IV at baseline, were observed in 91 patients (45%) and 57 patients (27%), respectively. In the whole cohort, the mean 6 MWT distance was 287 ± 168 m. The 6 MWT distance was significantly worse in patients with tissue ID compared to patients without tissue ID (206 ± 179 m vs. 314 ± 155 m, *p*-value < 0.0001) (Figure 1, Panel A). Likewise, an impaired submaximal exercise capacity was more common in patients with tissue ID (32, 64%) in comparison to patients without tissue ID (59, 39%, *p*-value = 0.002) (Figure 1, Panel B). Similarly, patients with submaximal impaired exercise capacity showed higher levels of sTfR (1.32 ± 0.66 mg/L vs. 1.53 ± 0.72 mg/L, *p* = 0.030) (Appendix A).

In the unadjusted GAM models (Figure 2) evaluating the interplay between the sTfR levels and the distance walked in the 6 MWT (submaximal exercise capacity), a significant linear association between increased iron demand (higher levels of sTfR indicating tissue ID) and worse submaximal exercise capacity (lower distance in the 6MWT) was seen.

Similar results were obtained in the regression models. As shown in Table 2, higher sTfR levels indicating increased iron demand were associated with a shorter distance in the 6 MWT (standardized β = −0.249, *p* < 0.001) and impaired exercise capacity (OR 12.2 (1.89–79.02), *p*-value = 0.009). Similarly, tissue ID was associated with a shorter distance in the 6 MWT (standardized β = −0.278, *p*-value < 0.001) and with an impaired exercise capacity (OR 2.8 (1.44–5.43), *p* = 0.002). These findings were confirmed in forward stepwise multivariate linear regression models (Table 2) and in multivariate GAM (parametric *p*-value < 0.0001) (Appendix A). Although the distribution of the NYHA functional class was different in the two groups (*p* = 0.020, Appendix A), the association between higher sTfR levels and tissue ID and an advanced NYHA class was not confirmed in the adjusted models (Table 2).

### 3.3. Association of sTfR with Quality of Life

To assess QoL, the Minnesota Living with Heart Failure Questionnaire (higher scores indicate worse QoL) was employed. In the whole cohort, the mean MLHFQ OSS was 42 ± 26 points (p). Overall QoL was significantly worse in patients with tissue ID compared to patients without tissue ID (51 ± 27 p vs. 39 ± 20 p, *p*-value = 0.006) (Appendix A).

In the unadjusted GAMs (Figure 3, Panel A) evaluating the interplay between the sTfR levels and the MLHFQ OSS (global QoL), a direct and significant association between increased iron demand (higher levels of sTfR indicating tissue ID) and worse overall QoL (higher scores in the MLHFQ OSS) was observed, with a linear *p*-value < 0.0001. Similar associations were observed with the physical and emotional dimensions of QoL (Figure 3, Panels B and C).

To confirm these associations, we developed several linear regression models (Table 3). In the unadjusted models, higher sTfR levels and tissue ID were associated with higher MLHFQ OSS scores, indicating worse global QoL. Similar results were observed for the physical and emotional domains of QoL. The association between the sTfR levels (standardized β = 0.183, *p*-value = 0.008) and tissue ID criteria (standardized β = 0.164, *p*-value = 0.019) and the MLHFQ OSS were confirmed in the adjusted linear regression models and were also consistent with the associations with the QoL physical and emotional domains (Table 3). Interestingly, the advanced NYHA class was the only additional independent predictor of a worse overall QoL in these models (standardized β = 0.195, *p*-value = 0.005). These adjusted associations were evaluated using a multivariate GAM (Appendix A) and were confirmed for overall QoL (MLHFQ OSS) (parametric *p*-value = 0.006) and the physical dimension scores of the MLHFQ (parametric *p*-value < 0.002).

## 4. Discussion

This study has shown that in patients with normal systemic iron parameters, higher levels of sTfR are strongly associated with an impaired submaximal exercise capacity. Moreover, it is confirmed that higher levels of sTfR are also correlated with worse QoL. The findings of the present work are very relevant and unique since this is the first time it has been shown that, in patients with HF without systemic ID according to the standard criteria, raised sTfR levels as a marker of increased iron demand may be indicative of an early stage of subtle iron insufficiency at the tissue level, promoting early impairments in submaximal exercise capacity and self-perceived health status.

ID is common in patients with HF, and it is associated with worse QoL, decreased exercise capacity, and worse outcomes regardless of the LVEF and hemoglobin levels [12,13,15,22,35,36]. Consequently, iron deficiency has been identified as a potentially modifiable factor that greatly influences functional status and QoL in patients with HF. In this regard, administering intravenous iron in patients with HF and ID significantly improved their symptoms, functional capacity, and QoL [8,9,37], reduced the risk of recurrent hospitalizations [38,39], and was shown to be safe in the long term in real-life populations [40].

This study adds new information regarding the impact of iron status and functional impairments in patients with HF. Particularly, we have shown that subclinical ID, defined as increased iron demand at the tissue level, may be regarded as an early stage in the continuum of iron status association with a negative impact on exercise capacity and QoL. The effects of this subtle iron insufficiency only affecting the functional compartment have been shown to occur despite normal parameters of the systemic iron status in terms of iron storage and transport in these patients. Moreover, multivariate analyses showed that sTfR levels (as a continuous measure of iron demand) and tissue ID (defined as sTfR levels in the upper quartile) were both independent determinants of the submaximal exercise capacity, impairment in exercise capacity, and overall, physical, and emotional QoL. Interestingly, a biomarker like sTfR allowed us to gauge the linear association between iron demand and functional measures in a continuous manner across the iron status continuum.

Regarding this, the criteria for diagnosing ID that are used in major clinical trials and recognized in both American [41] and European [3] HF guidelines have been questioned by several studies [42,43,44]. The current definition of ID in HF is serum ferritin < 100 ng/mL or TSAT < 20% [3,7]. This definition is based on the extrapolation of knowledge from the field of nephrology [45], but no formal validation of this definition has been performed in patients with HF [44]. The use of these parameters does have several limitations. These include the influence on ferritin levels by inflammation, infection, and malignancy. These conditions can falsely elevate the ferritin concentration regardless of the iron status [18]. In fact, the impact of the classic definition of ID on survival, adequately adjusted to co-morbidities and neurohormonal treatment in HF patients across the spectrum of LVEF, loses its predictive capacity [40,46]. Additionally, catabolism and malnutrition can reduce serum transferrin disproportionately to serum iron, leading to elevations in TSAT. In this setting, alternative definitions of ID have been proposed in recent years. They are based on several biomarkers, such as TSAT alone, serum iron, hepcidin, sTfR levels, or a combination of several of these.

Among them, the sTfR levels might be the most accurate. The sTfR concentration begins to rise early in iron deficiency with the onset of iron-deficient erythropoiesis, prior to the development of anemia [47,48]. This is why a rise in sTfR levels is a promising biomarker that could indicate subtle iron deficiency and a slight increase in iron demand at the tissue level, even in the absence of overt systemic iron deficiency or anemia.

High circulating sTfR has been identified as a strong independent predictor of long-term mortality in diabetic patients with coronary artery disease and a risk factor for myocardial infarction and cardiovascular death in patients with stable coronary artery disease [49,50]. Alcaide-Aldeano A et al. showed that sTfR levels were associated with impaired functional capacity and worse QoL in a patient cohort with systemic ID and HFpEF, [14]. However, no patients with a normal iron status (according to the standard criteria) or a reduced LVEF were included. Skikne BS suggested that sTfR levels accurately predicted depleted iron stores in the bone marrow of patients with ischemic HF. Additionally, it strongly predicted increased mortality in patients with ischemic HF with LVEF < 45% [21]. Thus, to the best of our knowledge, none of these groups have specifically studied patients with HF and normal iron status.

Our study presents original data since this is, to the best of our knowledge, the first study that confirms the previously mentioned associations between increased iron demand based on sTfR levels and submaximal exercise capacity, evaluated using the distance walked in the 6 MWT and QoL scores using the MLHFQ in patients with HF and otherwise normal hemoglobin and iron status at a systemic level, according to the standard criteria.

Recently, ID has been identified as a key element involved in the pathophysiology of HF and its progression, suggesting that ID might be more than a comorbidity [27,51]. Nevertheless, the mechanisms that lead to ID and mitochondrial dysfunction remain poorly characterized [52].

Normal iron metabolism is particularly critical for optimal cellular energy generation. Hence, ID primarily impairs the functioning of cells with a high energy demand (such as cardiomyocytes) [53,54]. This seems to be particularly important in the context of HF, as abnormal energy generation and utilization in the myocardium and the peripheral tissues (skeletal muscles) contribute to HF pathophysiology [55].

As shown in our study, we hypothesized that some patients with HF, despite normal iron status parameters in the storage and transport compartments, may suffer subtle forms of iron insufficiency at the tissue level. This represents a subclinical form of iron deficiency in the early stages of systemic ID. In these patients, raised sTfR values may represent an increased demand for iron at the tissue level that only affects cellular system concentrations and the activities of iron-dependent metabolism [17,51,55]. sTfR levels show the best correlation with myocardial iron content, and thus, could better indicate the iron status available to maintain cardiac function [56].

Although a deeper understanding of these pathways is required, the differential mechanisms regulating systemic and cellular iron and the unanswered questions in the pathophysiology underlying abnormal myocardial iron in HF should make us question the accuracy of the simple serum marker cutoffs used to diagnose ID in HF [57].

Our data suggest that a continuum approach instead of a cutoff value would be more accurate. This means that ID may be a continuum spectrum and not a dichotomic state in which the prognosis becomes worse as unmet iron demands rise.

Finally, our data support the use of QoL evaluation by means of the widely used MLHFQ tool, as it is also a validated questionnaire in the heart failure population. Additionally, we used the distance walked in the 6 MWT as an indicator of functional capacity. The 6 MWT is an easily performed, widely available, and well-tolerated test for assessing the functional capacity of patients with HF in everyday clinical practice. Although the gold standard for assessing functional capacity are maximal exercise tests, like the cardiopulmonary exercise test, previous studies have shown that the 6 MWT may provide reliable information about the patient’s daily activity and short-term prognosis [31,58].

There are many unanswered questions in this field that further research needs to clarify. These include the definitions of the pathways involved in the development of iron deficiency in HF and the potential use of sTfR and new potential biomarkers that would allow for better characterization of the iron status across its continuum.

Our study has several limitations that must be acknowledged. First, we conducted a cross-sectional analysis. Therefore, causality may not be inferred. In addition, this was a post-hoc analysis, whose primary endpoint was not the one for which the original study was planned. Thus, it is subject to all the biases linked to these kinds of sub-studies. Second, our findings may not be representative of other HF patient populations because this was a single-center study. Finally, the size of our sample was relatively small. With this consideration, larger studies are needed to validate these results.

## 5. Conclusions

In conclusion, we have shown that in patients with normal systemic iron parameters, higher levels of sTfR are strongly associated with an impaired submaximal exercise capacity and worse QoL. sTfR levels could be a good alternative and a more sensitive biomarker to define ID status, particularly in its early subclinical stages, since they could detect an early increase in iron demand at the tissue level.

## Figures and Tables

**Figure 1 jpm-13-01282-f001:**
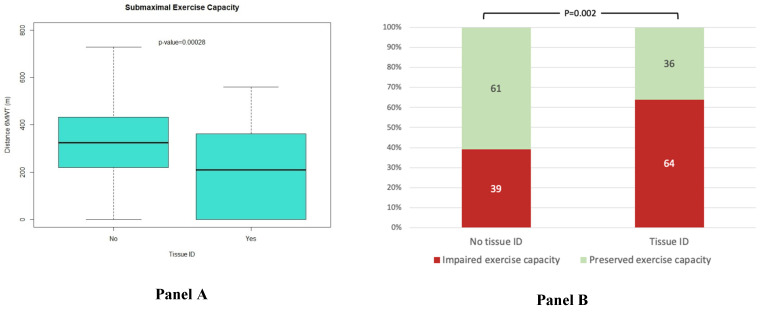
Panel (**A**). Boxplots showing the mean and standard deviation of the distance walked in the 6 MWT (meters) according to the presence of tissue ID (Tissue ID (+)) or absence (Tissue ID (−)). Panel (**B**). Proportions of patients with impairment in exercise capacity defined as 6 MWT distance < 300 m stratified by tissue ID status.

**Figure 2 jpm-13-01282-f002:**
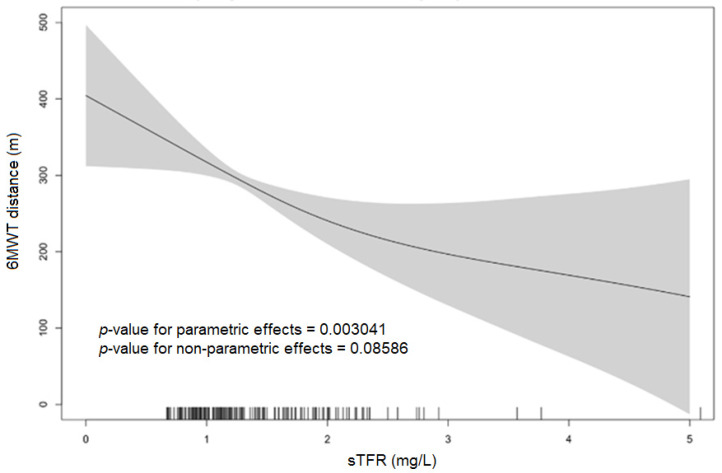
Univariate generalized additive models (GAMs) exploring the associations between sTfR levels and 6 MWT distance.

**Figure 3 jpm-13-01282-f003:**
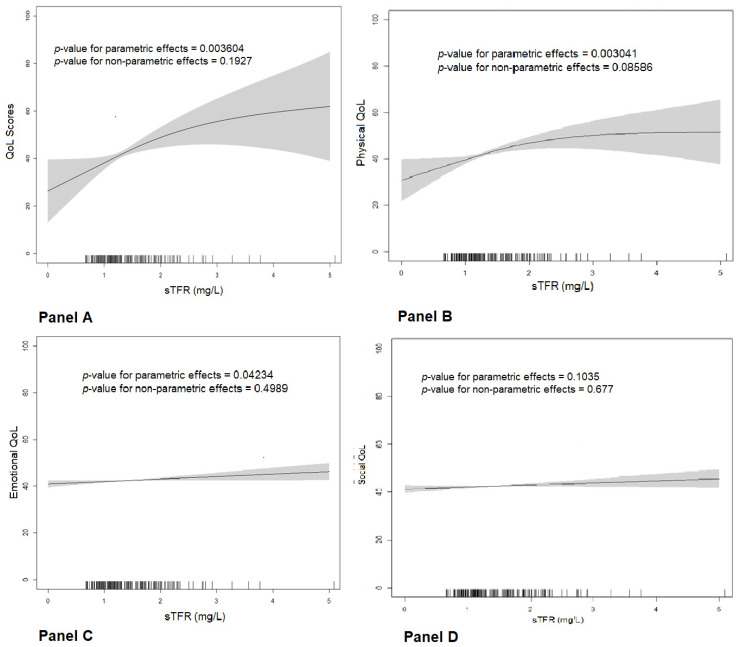
Univariate generalized additive models (GAMs) exploring the associations between sTfR levels and QoL scores (global QoL (Panel (**A**)), physical (Panel (**B**)), emotional (Panel (**C**)), and social (Panel (**D**)) dimensions).

**Table 1 jpm-13-01282-t001:** Demographic and clinical characteristics of all patients included in this analysis, overall and according to sTfR tertiles.

		Soluble Transferrin Receptor (sTfR) Levels, Divided into Tertiles			
		Lower Tertile	Middle Tertile	Upper Tertile	
	Whole Cohort (*n* = 215)	sTfR < 1.11 mg/L (*n* = 73)	sTfR 1.1–1.46 mg/L (*n* = 71)	sTfR ≥ 1.46 mg/L (*n* = 71)	*p*-Value
**Demographic and Clinical Factors**					
Age, years	70 (12)	68 (13)	70 (11)	72 (11)	0.113
Sex (female), *n* (%)	62 (29%)	23 (32%)	14 (20%)	25 (35%)	0.104
Systolic blood pressure, mmHg	125 (24)	124 (23)	127 (23)	124 (26)	0.807
Heart rate, bpm	73 (14)	71 (14)	72 (14)	77 (11)	0.020
NYHA Functional Class, *n* (%)	0.033				
I	44 (21%)	22 (31%)	15 (21%)	7 (10%)	
II	112 (53%)	36 (50%)	36 (51%)	40 (57%)	
III	46 (22%)	12 (18%)	18 (25%)	16 (23%)	
IV	11 (5%)	2 (3%)	2 (3%)	7 (10%)	
Impaired submaximal exercise capacity, *n* (%)	91 (45%)	25 (35%)	30 (46%)	36 (55%)	0.061
HF hospitalization in previous year, *n* (%)	172 (80%)	51 (71%)	63 (89%)	58 (82%)	0.025
LVEF, %	43 (15)	42 (15)	45 (14)	42 (15)	0.470
Ischemic etiology of HF, *n* (%)	64 (30%)	16 (22%)	23 (32%)	25 (35%)	0.183
**Comorbidities**					
Hypertension, *n* (%)	156 (73%)	50 (69%)	51 (72%)	55 (78%)	0.476
Diabetes mellitus, *n* (%)	68 (32%)	19 (26%)	28 (39%)	21 (29%)	0.202
Obesity (%)	58 (27%)	17 (23%)	21 (30%)	20 (28%)	0.671
Previous MI, *n* (%)	35 (16%)	8 (11%)	11 (16%)	16 (23%)	0.166
CKD, *n* (%)	90 (42%)	26 (36%)	31 (45%)	33 (47%)	0.361
**Treatments (%)**					
ACEI or ARBs	185 (86%)	64 (88%)	61 (86%)	60 (85%)	0.869
Beta-blockers	191 (89%)	64 (88%)	63 (89%)	64 (90%)	0.895
MRA	90 (42%)	34 (47%)	25 (35%)	31 (44%)	0.359
Diuretics	195 (91%)	62 (85%)	63 (89%)	70 (99%)	0.015
Antiplatelet or anticoagulant therapy	168 (78%)	50 (69%)	59 (83%)	59 (83%)	0.049
**Laboratory**					
Hemoglobin, g/dL	14.2 (1.4)	14.1 (1.3)	14.1 (1.4)	14.1 (1.5)	0.978
Creatinine, mg/dL	1.2 (0.4)	1.1 (0.3)	1.2 (0.3)	1.3 (0.4)	0.024
NT-proBNP, pg/mL (median, IQR)	1125 (587–2668)	1123 (505–2548)	966 (489–2144)	1660 (757–3871)	0.122
Serum proteins, g/dL	6.9 (0.7)	7.0 (0.8)	6.8 (0.7)	6.7 (0.7)	0.196
Serum albumin, g/dL	4.0 (0.6)	4.2 (0.5)	3.9 (0.5)	3.9 (0.7)	0.030

**Table 2 jpm-13-01282-t002:** Univariate and multivariate models exploring the effect on 6 MWT distance, impaired submaximal exercise capacity, and NYHA functional class limitations of sTfR and tissue ID in the cohort of non-anemic patients with HF and normal systemic iron parameters.

**Dependent Variable: Distance Walked in the 6 MWT (Submaximal Exercise Capacity) in Meters**				
	**Univariate Linear Regression Models**		**Multivariate Linear Regression Models**	
**Measures of Tissue ID**	**Standardized β Coefficient**	***p*-Value**	**Standardized β Coefficient**	***p*-Value**
Log sTfR (mg/L)	−0.249	<0.001	−0.135	0.010
sTfR > 75th percentile (1.63 mg/L)	−0.278	<0.001	−0.176	0.001
**Dependent variable: Advanced NYHA Functional Class (NYHA ≥ III or IV)**				
	**Univariate Binary Logistic Regression Models**		**Multivariate Binary Logistic Regression Models**	
**Measures of Tissue ID**	**OR (95% CI)**	***p*-Value**	**OR (95% CI)**	***p*-Value**
Log sTfR 1 (mg/L)	5.59 (0.852–36.671)	0.073	0.915 (0.079–10.594)	0.943
sTfR > 75th percentile (1.63 mg/L)	1.73 (0.887–3.368)	0.108	0.849 (0.364–1.980)	0.704

**Table 3 jpm-13-01282-t003:** Univariate and multivariate adjusted linear regression models exploring effects on quality-of-life scores (QoL) measured with the Minnesota Living with Heart Failure Questionnaire (higher scores indicate worse QoL) of sTfR levels and tissue ID in the cohort of non-anemic patients with HF and normal systemic iron parameters.

**MLHFQ Overall Summary Scores**					
	**Univariate Linear Regression Models**		**Multivariate Linear Regression Models**		
**Measures of Tissue ID**	**Standardized β Coefficient**	***p*-Value**	**Standardized β Coefficient**	***p*-Value**	**R Model**
sTfR (mg/L)	0.201	0.004	0.183	0.008	0.280
sTfR > 75th percentile (1.63 mg/L)	0.191	0.006	0.164	0.019	0.267
**MLHFQ Physical Dimension Scores**					
	**Univariate Linear Regression Models**		**Multivariate Linear Regression Models**		
**Measures of Tissue ID**	**Standardized β coefficient**	***p*-Value**	**Standardized β coefficient**	***p*-value**	**R model**
sTfR (mg/L)	0.204	0.003	0.176	0.012	0.310
sTfR > 75th percentile (1.63 mg/L)	0.213	0.002	0.151	0.029	0.377
**MLHFQ Emotional Dimension Scores**					
	**Univariate Linear Regression Models**		**Multivariate Linear Regression Models**		
**Measures of Tissue ID**	**Standardized β Coefficient**	***p*-Value**	**Standardized β Coefficient**	***p*-Value**	**R Model**
sTfR, 1 mg/L	0.142	0.042	0.132	0.057	0.258
sTfR > 75th percentile (1.63 mg/L)	0.161	0.021	0.148	0.034	0.258
**MLHFQ Social Dimension Scores**					
	**Univariate Linear Regression Models**		**Multivariate Linear Regression Models**		
**Measures of Tissue ID**	**Standardized β Coefficient**	***p*-Value**	**Standardized β Coefficient**	***p*-Value**	**R Model**
sTfR, 1 mg/L	0.114	0.102	0.132	0.052	0.305
sTfR > 75th percentile (1.63 mg/L)	0.061	0.384	0.096	0.162	0.305

## Data Availability

No new data were created or analyzed in this study. Data sharing is not applicable to this article.

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
