# Peer review of "Soluble Transferrin Receptor as Iron Deficiency Biomarker: Impact on Exercise Capacity in Heart Failure Patients"

_jpm, 2023, doi:10.3390/jpm13081282_

Round 1

Reviewer 1 Report

As several papers highlight, cardiovascular disease is the leading global cause of death, and most of the surviving patients require screening, preventative care, and coordinated follow-up appointments because of their increased risk of developing heart failure. Biomarkers are critical tools for probing, assessing, and managing cardiovascular risk. The state-of-the-art biomarkers accurately diagnose acute CVD but are limited to predicting CVD progression. In this manuscript, Ras-Jimenez et al., propose using sTfR, in correlation with QoL questionnaire, as a parameter to detect ID at the tissue level and the progression of heart failure. This is a promising study in new biomarkers for cardiovascular disease.

Major review:

  1. The authors explained several methodological details with few results/discussion in the abstract.
  2. The introduction must be revisited. Paragraphs were not clear, short, and some references were missing (e.g., Paragraph 7, page 2, lines 93-95).
  3. The comparison between TSAT vs. sTfR is not clear in the introduction.
  4. The last 2 paragraphs of the introduction are redundant, making the authors' hypothesis weak.
  5. The reviewer suggests a total reformatting of the Material and Methods section, separating it into subtopics. E.g., Patient inclusion criteria, Iron deficiency by sTfR, and statistical analyses.
  6. Reading the methods, the reviewer understood that the authors used the data collected during the DAMOCLES clinical trial. So, this manuscript is an ancillary study, meaning that most of the information on page 3, lines 130-140 was unnecessary.
  7. On page 3, lines 147-149, the sentence looks like a paragraph from the introduction; please clarify. Same for page 4, lines156-157.
  8.  Not clear if the sTfR assay was performed during the clinical trial or just for this study.
  9. The reviewer could not open the supplemental data once the file was corrupted.
  10.  The result section was clear and well presented. The authors must fix the figure legends (bottom of each image) and the wrong location over the text.
  11.  Although the discussion section presents comprehensive literature, it lacks a few comparisons and critical analysis of the authors' findings.

Minor:

  1. Check MDPI guidelines for how to present the authors' names.
  2. Some spelling mistakes in the text
  3. Fix the keywords' punctuation
  4. Use abbreviations. E.g., Page 2, line 87, quality of life was already presented as QoL, and lines 93-94, iron deficiency, as ID.
  5. On page 2, line 90, “%” transferrin saturation (TSAT) is redundant.

Minor English editing

Reviewer 2 Report

Formatting issue in the Title – Please fix

Abstract:  Consider reformatting it with the standard subject, methods, results, and conclusion format since it would be more appropriate.

Introduction – when you say several studies multiple times. Consider including qualitative data and more details when you write like “these studies”, “several clinical trials”, “this idea”, and “these studies” followed by a claim statement. I,e., line 82-87.  What studies? What clinical trials? Please be specific.

If you are making a statement or claim, please back it up with specific qualitative data from the referenced studies. THIS IS TO FIX THROUGHOUT THE DOCUMENT.

Include references to guidelines when using them to support your claim such as European Society  of Cardiology Diagnostic criteria. Please include references.

Some sentence structure needs, reformatting. (165-166). Spacing in sections 3.1, 3.2 and other places are off. Please see the formatting of this script.

The discussion looks great, with no further major comments. Authors to consider including more references from other larger studies and registry data involving iron deficiency and heart failure and post-HF 6MWT data (which is publicly available) should be compared and analyzed to strengthen the claim. However, for this sample size, the outcomes are acceptable.  

Formatting issue in the Title – Please fix

Consider reformatting it with the standard subject, methods, results, and conclusion format since it would be more appropriate.

Some sentence structure needs, reformatting. (165-166). Spacing in sections 3.1, 3.2,

Round 2

Reviewer 1 Report

The authors addressed all reviewer’s concerns regarding the study, substantially improving the text quality and the images. Because the reviewers didn’t receive a clear version of the text, It’s hard to detect minor issues. I believe the editorial management of MDPI can work directly with the authors to fix any minor errors if they exist.